# Vitamin D as a Shield against Aging

**DOI:** 10.3390/ijms24054546

**Published:** 2023-02-25

**Authors:** Cristina Fantini, Clarissa Corinaldesi, Andrea Lenzi, Silvia Migliaccio, Clara Crescioli

**Affiliations:** 1Department of Movement, Human and Health Sciences, University of Rome “Foro Italico”, Piazza Lauro de Bosis, 006-00135 Rome, Italy; 2Institute for Cancer Genetics, Columbia University, New York, NY 10027, USA; 3Department of Experimental Medicine, Sapienza University of Rome, Viale del Policlinico, 155-00161 Rome, Italy

**Keywords:** vitamin D, immunosenescence, inflammaging, molecular mechanisms, immunocytes, cardiomyocytes, skeletal muscle cells

## Abstract

Aging can be seen as a physiological progression of biomolecular damage and the accumulation of defective cellular components, which trigger and amplify the process, toward whole-body function weakening. Senescence initiates at the cellular level and consists in an inability to maintain homeostasis, characterized by the overexpression/aberrant expression of inflammatory/immune/stress responses. Aging is associated with significant modifications in immune system cells, toward a decline in immunosurveillance, which, in turn, leads to chronic elevation of inflammation/oxidative stress, increasing the risk of (co)morbidities. Albeit aging is a natural and unavoidable process, it can be regulated by some factors, like lifestyle and diet. Nutrition, indeed, tackles the mechanisms underlying molecular/cellular aging. Many micronutrients, i.e., vitamins and elements, can impact cell function. This review focuses on the role exerted by vitamin D in geroprotection, based on its ability to shape cellular/intracellular processes and drive the immune response toward immune protection against infections and age-related diseases. To this aim, the main biomolecular paths underlying immunosenescence and inflammaging are identified as biotargets of vitamin D. Topics such as heart and skeletal muscle cell function/dysfunction, depending on vitamin D status, are addressed, with comments on hypovitaminosis D correction by food and supplementation. Albeit research has progressed, still limitations exist in translating knowledge into clinical practice, making it necessary to focus attention on the role of vitamin D in aging, especially considering the growing number of older individuals.

## 1. Introduction

The undeniable increase observed in human life expectancy and longevity in developed countries is too often accompanied by a decrease in the quality of life (QoL) [1]. This means that living longer is not living better, thus, the current challenge is how to maintain and improve a good QoL during a longer lifespan. The aging process is a gradual functional decline, that starts early in adulthood, continues throughout the human lifespan, and might end in chronic diseases, e.g., sarcopenia, metabolic disorders, cardiovascular and neurodegenerative diseases, or cancer [2]. These macroscopic age-related alterations reflect the changes occurring at the cellular and molecular levels: mitochondrial dysfunction, damaged protein accumulation, epigenetic alterations, telomere shortening, aberrant intracellular signaling, and altered nutrient sensing are recognized to be the biomolecular pillars of aging [3,4]. Altogether, these molecular alterations converge in frailty, a multidimensional syndrome, clinically marked by decreased physiological reserve and resistance to stressors, leading to a higher rate of disability and mortality [5]. Frailty is undeniably a multifaceted condition, due to different determinants, however, the reduced capacity in immune defense, known as “immunosenescence”, and the consequent increase in chronic low-grade inflammation, named “inflammaging”, emerge as the main triggers. Whereas aging is a natural and unstoppable process, several adaptable lifestyle factors can favorably impact QoL, by strengthening the immune system and promoting anti-inflammatory processes.

It is well-known that diet can drive human health status toward wellbeing or disease. Indeed, the nutritional pattern per se plays a pivotal role in promoting healthy aging, by modulating several intracellular processes.

Among many factors, vitamin D plays an important role as a nutrient capable of affecting the aging process at a cellular/molecular level, with a wide range of actions.

Indeed, beside bone defects, insufficient vitamin D is associated with the increased risk of developing a wide range of pathologies, from neurological diseases to cancer, obesity, diabetes and metabolic disturbance, cardiovascular diseases, autoimmune diseases, infections, and menopause-related diseases [6,7,8,9,10,11,12,13].

A clear cause–effect relationship still lacks in many cases, nevertheless hypovitaminosis D is a marker for poor health, especially in the elderly, often characterized by co-morbid conditions [14].

Most of the actions of vitamin D are undoubtedly ascribable to its fine-tuned immunomodulatory properties, impacting immune system function and inflammation. This review aims to provide an overview of some of the cellular biomechanisms and biomolecules engaged in immunosenescence and inflammaging, as biotargets for vitamin D interventions during adult life, or even earlier, in order to maintain a level of health as high as possible while aging. Particular attention will be given to the ability of vitamin D to shape the human immune status, acting on the immune system or immune-related activity of some tissues, such as skeletal muscle or heart, whose function highly affects QoL during life, including in old age.

## 2. Aging Is a Cellular and Molecular Matter 

Human aging is a complex process influenced by several factors, including sex, genetics, socioeconomic status, and lifestyle. Age-related changes are associated with higher susceptibility to diseases, depending on macroscopic modifications in the whole-body physiology and organ function, which, in turn, mirror the changes occurring at the cellular and intracellular levels. Accumulating data suggest that dietary pattern, among other factors, can prevent some molecular/cellular changes underlying human aging-related diseases, and maintain the functional abilities associated to wellbeing [World Health Organization Ageing: Healthy Ageing and Functional Ability 2020; https://www.who.int/news-room/questions-and-answers/item/healthy-ageing-and-functional-ability, (accessed on 26 October 2020)]. Some of the main processes triggering and mediating senescence and inflammation are discussed below.

### 2.1. The Cellular Route to Aging

Understanding the molecular drivers of age-dependent multimorbidity can help to develop new interventions and strategies to delay this condition [15]. 

The multimodal process of aging includes different biological system remodeling leading to the loss of homeostasis and, consequently, deterioration of several organs and tissues. In this scenario, age-related biomolecular changes in the immune response and immune cells are acknowledged to play a causal role in driving whole-body aging [16]. In humans, the natural decline in immune system function usually starts in the sixth decade, and tends to continuously progress toward immunosenescence and decreased protection against pathogens; at the same time, the inflammatory response increases, in terms of duration/intensity [17]. Age-dependent “re-shaping” of both the innate and adaptive arms of the immune landscape can increase vulnerability to illness, and promote and allow multimorbidity, frailty, and adverse health outcomes [18]. 

Indeed, whereas the adaptive immunity decreases and leads to weakened antigen-specific response and impaired memory formation, the nonspecific innate immunity overreacts, leaving older individuals unprotected from chronic inflammation. The concurrence of these conditions is defined as “inflammaging” [19,20]. In turn, inflammaging is an additional risk factor in the elderly, since it amplifies other age-associated modifications related to morbidities.

The macroscopic changes reflect microscopic remodeling at the cellular and subcellular levels, involving significant modifications in intracellular paths, especially in the T cell repertoire [21,22,23].

Cellular senescence combines morphological and molecular characteristics, makes cells modify cyclin-dependent kinases (CDKs) regulation, and undergo permanent cell cycle arrest, in a steady status different from terminal differentiation or quiescence [24,25]. The diluted cytoplasmic domain, the enlarged cell size, and increased β-galactosidase lysosomal activity—mostly used as a senescence cell marker—are the major features of senescent cells, along with an altered DNA and chromatin landscape, a damaged protein and lipid profile, and dysfunctional mitochondria and lysosomes [24,26,27]. Remarkably, senescent cells display a “senescence-associated secretory phenotype” (SASP) due to a specific secretory profile or “secretome”, which is characterized by an excessive secretion of proinflammatory molecules, such as cytokines and chemokines, matrix metalloproteinases, and growth and angiogenic modulators [28,29,30]. Figure 1 shows a schematic of the cellular changes associated with senescence.

In this scenario, the production and release of proinflammatory molecules increase, including interleukin (IL)-6, IL-1, tumor necrosis factor(TNF)α, interferon (IFN)γ, acute-phase proteins, reactive oxygen species (ROS), and autoantibodies constantly increase and accumulate. These factors function as mediators of signal communication between cells resident within organs/tissues, and immune cells, and thus amplify inflammaging [31,32].

To date, several in vitro and in vivo studies show that the senescence-related signature is highly variable, but a robust approach to establishing a senescent secretome profile is still lacking [33]. Nevertheless, senescence of the immunocytes is undeniably acknowledged as a critical step in triggering sustained chronic-persistent inflammation, therefore, representing a potential target for interventions.

### 2.2. Immunosenescence: The First Step toward Aging

The central defect in immunosenescence is the decline in T cell function, as shown by animal and human studies, leading to overactivity, inflammation, and autoantibody production, with quite predictable consequences on health [34,35,36,37].

The main age-dependent changes include a significant decrease in naïve T cells, along with a simultaneous increase in memory cells, impairments in the T cell receptor (TCR) repertoire, defects in NK cells and neutrophil chemotaxis, myeloid skewing, and monocyte dysregulation. Albeit immunosenescence involves most of the immunity-related components, T cells emerge as playing a triggering role in aging, as shown by investigations on rheumatoid arthritis (RA), an autoimmune disease in which the faster progression of this process allowed the characterization of some underlying molecular paths [38,39,40].

Age-related thymus involution is the first event that triggers impairments and defects in T cell differentiation and maturation. The number of new naïve T cells decreases, and circulating naïve T cells live longer, accumulate defects, and cause a decreased TCR diversity [34]. T cell-TCR signaling activation is required for IL-2 release and T cell expansion, which are known to be critical for mounting an effective immune response [41]. In aging, the downtuning of T regulatory (Treg) cells leads to an unbalanced Th17/Treg ratio, with Th17 upregulation and Treg decrease, ending in polarization of the immune response towards inflammation [42].

Furthermore, age-related thymic involution, naïve T cell dysfunction, and aberrant expression of age-associated gene profiles seem to be linked to IL-33/ST2 signaling activation, suggesting that targeting IL-33 or ST2 could be a promising strategy to rejuvenate T cell immunity [43]. IL-33 has been shown to orchestrate the signals within the skeletal muscle/immune system/nervous system in response to injury. Remarkably, the IL-33/ST2 axis is suggested as a promising potential target in managing age-related sarcopenia and muscle repair due to injury or atrophy, both major problems impacting mobility and QoL in the elderly [44]. 

The dysregulation of T cell signaling includes defects in calcium mobilization, phosphorylation of tyrosine and serine/threonine, mitogen-activated protein kinases (MAPK) activity, and activation of transcription factors such as nuclear factor kB (NF-kB), nuclear factor of activated T cells (NFAT), or activator protein 1 (AP-1) [45,46,47,48,49,50,51,52].

Other age-dependent damage, i.e., the loss of co-stimulatory molecule CD28—which plays a pivotal role in cell activation/proliferation/survival—impacts on the function of B cells (proliferation and Ig production) and the antigenspecific cytotoxic CD8 T cell subset [34].

In addition to T cell deregulation, aberrant increases in NF-kB activation and cyclooxygenase (COX)-2 expression, with a higher production of prostaglandin E2 (PGE2) from macrophages, have been reported in the elderly [53,54,55].

Remarkably, NF-kB dysregulation/hyperactivation represents a potential molecular target for intervention, with particular significance in the elderly, as addressed later in this review.

The number of natural killer (NK) cells, which are known to have cytotoxic/lytic activity against cancers and viruses, seems to not vary with aging, whereas cytokine and chemokine production declines [56,57,58]. Like NK, neutrophil number does not change with age, but neutrophil activity, including oxidative burst, phagocytosis, and chemotaxis (all defense mechanisms) is significantly compromised [57].

In the complex network of aging, oxidative stress is highly determinant, as suggested by the coined term “oxi-inflamm-aging” [59]. Indeed, the high percentage of polyunsaturated fatty acids present in immune cell plasma makes the cells inclined to lipid peroxidation; any damage in the oxidative process can alter signal transmission within/between several types of immunocytes, ending in a defective immune response [60,61].

Thus far, age-related changes in the immune system expose the elderly to a higher risk of disease, first of all, cancer, and infections.

The significant impact of genetic and environmental factors on immune system function is undeniable, and nutrition emerges as an acknowledged tool to regulate the immune status. Indeed, age-related immune alterations seem to be associated with a suboptimal nutrient status, so that nutritional interventions with macronutrients (i.e., polyunsatured fatty acids or PUFA) or micronutrients (i.e., vitamins, elements) are highly recommended for ameliorating QoL in the elderly. Whereas macronutrients provide a substrate for the biosynthesis of molecules engaged in immune response (acute-phase proteins, cytokines, new receptors, amino acids for immunoglobulins), and are fuel for immune cell energy, micronutrients, such as zinc, iron, and vitamins are exquisite regulators of the immune response at the cellular and molecular level [62].

Overall, this review focuses on vitamin D as a fine-tuned regulator of both the innate and adaptive immune responses, playing a pivotal role in the health of individuals of any age, particularly, in the elderly.

## 3. Vitamin D and Geroprotection

Vitamin D has historically been known as the necessary nutrient to guarantee a correct bone metabolism and health. This molecule is classically defined as a steroid hormone, since it shares with steroids the common progenitor molecule (cyclopentanoperhydrophenanthrene). It is synthesized as the precursor molecule in skin exposed to sufficient UV rays, and is transformed into the biologically active compound by two enzymatic hydroxylations in the liver and kidneys (25- and 1-α-hydroxylase, respectively). Therefore, while “vitamin” is considered an imprecise term [63], this molecule is present in food such as fatty fish, i.e., salmon and trout, beef liver, cheese, egg yolk, albeit the latter ones provide small amounts, mushrooms and some vegetables, and is fully identified as an essential micronutrient with critical regulatory functions [64].

Vitamin D deficiency is a worldwide problem, affecting people of all ages, due to several impacting factors, from ethnicity and skin color, to latitude, habits and lifestyle, sex, and age. It has been reported that the elderly in Europe, the USA, and Australia suffer from insufficient levels of vitamin D, mainly due to less solar exposure, associated with low levels of outdoor activity or with clothing, and also due to reduced synthesis, due to atrophic skin modifications and a lower amount of precursor or reduced renal function [65,66]. Remarkably, a less varied diet, with a lower vitamin D content, is a typical age-associated habit, contributing to vitamin D deficiency in aging populations [67,68].

Thus far, vitamin D intake to a sufficient level is recommended not only for bone homeostasis but to maintain a general status of good heath as well, especially during aging, considering its pleiotropic effects. Vitamin D intake from dietary sources, in particular foods enriched or fortified with this nutrient, is considered an excellent strategy to counteract vitamin D deficiency, a condition currently defined as being a “world problem” [69].

It is acknowledged that this nutrient exerts such an important effect on the immune response that it can be considered a tool to tackle immunosenescence and oxi-inflamm-aging. In general, vitamin D deficiency is associated with a higher risk of infections and autoimmune diseases, involving dysfunctional biological activity of the specific vitamin D receptor (VDR), which is expressed in the majority of immunocytes [70]. Hypovitaminosis D is acknowledged as a pandemic condition, affecting all stages of life, with detrimental consequences on health; but in the elderly, vitamin D deficiency retains even more clinical significance, since this condition often converges with other age-related deficiencies (i.e., hormonal) or diseases, worsening the outcome.

Most of the immune cells, including T and B cells, dendritic cells (DC), macrophages and monocytes, express VDR and respond to vitamin D with fine-tuned modulations in cell signaling, path activation, and molecule production, with significant consequences on immune response [71,72,73,74,75,76,77,78].

Furthermore, many immunocytes express 1-α-hydroxylase and can themselves produce the active metabolite and, thus, control the local cell microenvironment [79,80].

In the elderly, adequate vitamin D levels help to counteract the natural decline in immune surveillance by a fine-tuned orchestration of several effects.

### 3.1. Vitamin D against Infections

Vitamin D strengthens the first line of host defense, that is particularly relevant during aging, when the risk of infection is higher. Indeed, this molecule can maintain the barrier integrity and induce a set of genes encoding antimicrobial proteins (AMPs), such as cathelicidin, defensins, hepcidin, and neutrophil peptides, which behave as antibiotics against various types of pathogens [81,82].

The vitamin D–cathelicidin axis is the most studied and best characterized among the VDR-dependent signaling engaged in infections and autophagosome formation—the latter one playing a critical role in microorganism clearance and infection resolution. Clinical studies on human tuberculosis, sepsis, viral infection, peritonitis, and pneumonia, whose incidence rise in the elderly [83], document that, after vitamin D supplementation, serum cathelicidin—human cathelicidin LL-37 or human cationic AMP 18 (hCAP-18)—increases, and correlates with improved clinical outcomes [84,85,86].

Evidence in human monocytes/macrophages shows that the activation of vitamin D signaling, triggered by toll-like receptor (TLR)2/1 or TLR8, leads to an increase in antimicrobial response, autophagy, antimicrobial peptide expression, and phagosome–lysosome fusion, in association with an increase in IFNγ, which is likely required to strengthen antibacterial activity, together with IL-12 and IL-18 [87,88,89,90].

Furthermore, IFNγ expression, in combination with CD40–CD40 ligand signaling, increases the activity of the hydroxylase converting 25-hydroxyvitamin D (25D) to the active metabolite, in human monocytes [91].

In Mycobacterium tuberculosis, the most studied infection, vitamin D-induced cathelicidin acts as a second messenger to activate autophagy genes, such as autophagy related 5 (ATG5) and beclin-1 (BECN1), and triggers a downstream wide signaling cascade, including intracellular calcium (Ca^2+^) release, Ca^2+^-dependent kinases, extracellular ATP-gated ion channel, purinergic receptor (P2X) 7, mammalian target of rapamycin (mTOR)/AMP-activated protein kinase (AMPK)/phosphoinositide 3-kinase(PI3K) pathway, and reactive ROS signaling [92,93,94].

During Mycobacterium infection, vitamin D boosts cytokines/chemokines production through the induction of IL-1β, that regulates defensin beta 4 gene (DEFB4), encoding for human beta-defensin-2 (HBD2) in macrophages [95]. 

Thus far, vitamin D/VDR signaling acts as a fine regulator of AMP-dependent autophagy, cytokines/chemokines production, IFN-dependent signaling, and ROS generation [71,96,97].

Since ROS mediates TLR2-induced cathelicidin expression in human monocytes/macrophages [98], and ROS-autophagy events are mutually regulated [99], further understanding the function of the vitamin D–cathelicidin axis in redox homeostasis and autophagy activity seems necessary. The antimicrobial action(s) of host defense proteins induced by vitamin D might be particularly relevant, since infections from drug-resistant pathogens are emerging worldwide [100]. 

Remarkably, vitamin D-mediated LL-37 induction is specific in human defense system, as this vitamin D function has been reported to fail in a murine system [101]. This observation is in line with the concept that, results from animals are often not translatable to humans, especially concerning studies on immunity [96].

However, the specific role of the vitamin D–cathelicidin axis in the different infections caused by bacteria, viruses, and parasites goes beyond the aim of this paper, and was exhaustively reported in another recent paper [88]. 

To date, whereas inflammatory cytokines such as TNF-α and IL-1β are central within vitamin D-mediated antibacterial activity during host defense, alterations in the vitamin D–cathelicidin axis, leading to lower vitamin D and excessive cathelicidin, result in overinflammation, and take part in the pathological setting of chronic inflammatory diseases, such as rosacea, a disease whose prevalence increases with age [102,103]. 

The paradoxical effect of vitamin D emerges; while this molecule works on the one hand to keep active the macrophage phenotype (M1), associated with proinflammatory/anticancer activity, to preserve physiological integrity, on the other hand, it helps to maintain the pro-tolerogenic/anti-inflammatory phenotype (M2), to counteract inflammaging, that is the route opening to a variety of diseases. The main anti-inflammaging effects related to vitamin D are addressed in the following paragraph. Figure 2 depicts the effects of vitamin D against inflammation, as well as some other molecular effects, reported in the following paragraphs.

### 3.2. Vitamin D against Inflammaging, Cellular Senescence, and Mitochondrial Dysfunction 

Following the original concept, inflammaging is a consequence of immunosenescence, even if this model has been re-interpreted in favor of a mutual interplay [104]. 

The immunomodulatory role of vitamin D in inflammation is widely recognized. Substantially, an adequate level of vitamin D counteracts inflammation with multilevel targeting effects, i.e., inhibiting the expression and signaling of TLR2, 4, and 9, reducing the production of cytokines such as TNF-α, IL-6, IL-23, and IL-1, and repressing the activity of T cells recruiting chemokines [105].

The main inhibitory effects are on CD4+ and CD8+ T cell proliferation, in particular T helper 1 (Th1) cells (a subset of CD4+ effector T cells), in which vitamin D counteracts the release of cytokines, i.e., IL-2 or IFNγ, capable of activating macrophages, and IL-6, IL-8, IL-12, and TNFα, all molecules characterizing inflammaging [106,107]. 

The vitamin D-induced downregulation of the proinflammatory Th17 cell subset, suppression of dendritic cell (DC) maturation from monocytes, and impaired capacity to present antigens, occur together with Treg enhancement and an increase in anti-inflammatory cytokines such as IL-4, IL-5, IL-10, and CCL2. Altogether, these processes are supposed to be the main mechanisms underlying Th2 protolerogenic subset expansion, which, in turn, is able to mitigate inflammaging and autoimmune disorders [108,109,110,111,112,113].

In lymphocytes, vitamin D has been documented to inhibit IL-6, a critical factor in stimulating the Th17 cell subset, which plays a pivotal role in autoimmune reactions [114]. 

From in vitro studies on autoimmune diseases, the vitamin D-induced protolerogenic effect on T cells seems undeniable, but less effectiveness in vivo is hypothesized, maybe related to a different ability of T cells to modify their phenotype in response to vitamin D (more phenotypically committed, less responding to vitamin D) [115]. The precise biomolecular mechanism remains to be elucidated. Albeit the etiology of autoimmune diseases is complex and undeniably multifactorial, the role of hypovitaminosis D is acknowledged as highly impacting disease development [116].

In some autoimmune diseases, such as osteoarthritis, psoriasis-associated osteoporosis, and Guillain–Barré syndrome, a pathogenetic immunological crosstalk between vitamin D and IL-33 may be hypothesized, with the combination of hypovitaminosis D–IL-33/ST2 axis activation converging in deleterious effects. Hence, the neutralization of IL-33/ST2 signaling by vitamin D in immunocytes (T cells, DC) is suggested as a therapeutic strategy [117,118,119].

In this scenario, the Th17/Treg ratio tends to decrease simultaneously with a significant increase in the transcription factor fork head box (Fox)P3, the most reliable marker to date for Treg cells, and IL-10 [120]. 

Remarkably, IL-10 is the anti-inflammatory cytokine known to keep inflammaging and antigenic stress under control, thus representing a good molecular defense mechanism in the elderly [121]. In line with this concept, the poorer vitamin D status—i.e., due to reduced sunlight exposure, declined ability of the skin to produce vitamin D, malnourishment, or decreased vitamin D intake—observed in elderly people, likely contributes to the higher prevalence of a variety of age-related diseases associated with a compromised immune system [122,123,124,125]. As shown in COVID-19 patients, a wide-epigenetic T cell remodeling, promoting VDR expression and enzyme cytochrome P450, family 27, subfamily B, member 1 (CYP27B1) activation in autocrine/paracrine mode, likely underlies the transition from proinflammatory IFN-γ+ Th1 cells (via STAT3, c-JUN, and BACH2) to suppressive IL-10+ cells (via IL-6–STAT3 signaling) [126,127]. In addition to the effect on immunocytes, vitamin D deficiency increases endothelial senescence, allowing vascular dysfunction and atherosclerosis, both processes that increase in prevalence with aging.

This effect is undeniably related to vitamin D’s anti-inflammatory action, as inflammation is a recognized trigger of atherosclerosis initiation, progression, and plaque and thrombus formation, also in the young [128]. However, the vitamin D-induced reduction in cholesterol uptake by macrophages and, in turn, suppression of foam-cell formation, emerges as a key event [129].

So far, hypovitaminosis D supports increased cellular senescence and arterial aging, characterized by the typical dysfunctions as a gradual loss of vascular smooth muscle cells’ contractility, and increased arterial permeability and intima thickness [130]. 

The delayed cellular senescence by vitamin D/VDR interaction includes different processes, including longer telomere lengths, an increased antioxidant effect via nuclear factor erythroid 2-related factor 2 (Nrf2) transcriptional regulation, decreased oxidative stress, DNA damage, and SASP, downregulation of p16, p53, and p21 (major regulators of the G1/S cell-cycle checkpoint), while upregulating Bmi1(polycomb ring finger oncogene, transcriptional suppressor), whose novel action is reported in cardiac regulation [131,132].

Another important aspect is the regulation of mitochondrial function. Vitamin D deficiency is associated with disorders of mitochondrial function, such as respiratory chain deregulation, with downregulation of mRNA and proteins involved in mitochondrial respiration, inhibition of sirtuin (SIRT) 1, which plays a pivotal role in mitochondrial biogenesis through PGC-1α, and, e.g., in brain aging delay [133,134].

The combination of vitamin D and curcumin, given as a supplement in a continuous way, has been recently hypothesized to counteract neurodegeneration [135]. SIRT signaling and lifespan seem deeply affected by dietary nutrient composition and resveratrol, which suppresses oxidant and inflammatory genes, altering promoter epigenetic status. The immune system is key to a host’s defense against pathogenic organisms. Aging is associated with changes in the immune system, with a decline in protective components (immunosenescence), increased susceptibility to infectious disease, and a chronic elevation in low-grade inflammation (inflammaging), increasing the risk of multiple noncommunicable diseases. Nutrition is a determinant of immune cell function and of the gut microbiota. In turn, the gut microbiota shapes and controls the immune and inflammatory responses. Many older people show changes in their gut microbiota. Age-related changes in immune competence, low-grade inflammation, and gut dysbiosis may be interlinked, and may relate, at least in part, to age-related changes in nutrition. A number of micronutrients (vitamins C, D, and E, and zinc and selenium) play roles in supporting the functions of many immune cell types. Some trials have reported that providing these micronutrients as individual supplements can reverse immune deficits in older people, and/or in those with insufficient intakes. There is inconsistent evidence that this will reduce the risk or severity of infections, including respiratory infections. Probiotic, prebiotic, or synbiotic strategies, that modulate the gut microbiota, especially by promoting the colonization of lactobacilli and bifidobacteria, have been demonstrated to modulate some immune and inflammatory biomarkers in older people, and in some cases, to reduce the risk and severity of gastrointestinal and respiratory infections, although, again, the evidence is inconsistent. Further research, with well-designed and well-powered trials, in at-risk older populations is required to be more certain about the role of micronutrients and of strategies that modify the gut microbiota–host relationship in protecting against infection, especially respiratory infection [136,137]. COX-2 expression has been shown to be dose-dependently inhibited by vitamin D in murine macrophages, through suppression of the protein kinase B (PKB) or Akt/NF-kB/COX-2 pathway [138]. 

The function of NF-kB is critical in cellular senescence and in inflammaging, since this factor is suggested to serve as a linkage between aging hallmarks in cell–cell communication, and in age-related pathophysiological mechanisms [139,140]. Indeed, while NF-κB unaltered signaling supports the correct interplay between immunocytes and non-immune cells, to maintain a functional host response, NF-kB hyperphosphorylation and enhanced activity are reported in different tissue aging (skin, hypothalamus, and cortical tissues) [141,142]. Similarly, NF-kB constitutive activation is reported in aged skeletal muscle [143].

To date, in vivo data in overweight/obese, but otherwise healthy, individuals failed to show any significant difference in NF-kB activity; however, the inadequacy of sample size to detect such a difference as a primary outcome is indicated by the authors as a possible reason for their null findings [144]. Nevertheless, the function of NF-kB as a regulator of inflammation and immunity continues to emerge as being critical for aging and age-related diseases [145]. NF-kB signaling hyperactivation is associated with pro-aging stimuli, and correlates with the development/progression of most aging-related diseases, whereas NF-kB inhibition can delay or even reverse aging processes [146,147,148,149].

Thus far, inflammaging likely primes inflammatory NF-kB signaling, which is involved in a variety of age-related tissue/organ dysfunctions. Interestingly, vitamin D can target hyperphosphorylation of this transcription factor in several cell types. The following paragraph focuses on this topic.

## 4. Vitamin D-Dependent NF-kB Regulation: A Putative Anti-Aging Strategy

NF-kB can be considered as the central crossroad where many paths and signals, either promoting or delaying aging, converge, by activating or inhibiting this transcriptional factor, respectively [147]. Biomolecular paths promoting aging such as insulin (I)/insulin growth factor (IGF)-1 signaling, activate NF-kB via the PI3K/AKT cascade and mTOR, a known pro-aging factor [146,150,151,152]. In addition, age-dependent DNA damage and telomere shortening are associated with NF-kB aberrant activation, and with increased levels of COX-2 and ROS [153]. NF-kB activation, besides telomere shortening via telomerase reverse transcriptase (TERT) catalytic subunit activation, intensifies inflammation by macrophage polarization to M1 phenotype, and IL-6 and TNFα upregulation [154,155], and increases SASP and cellular senescence [147]. At variance to this, pro-longevity factors, i.e., sirtuin and FOXO, repress NF-kB transcriptional activity, by directly interacting with the p65 subunit [156,157,158]. To date, chronic activation of NF-kB is found in several age-related diseases, e.g., atherosclerosis, osteoporosis, muscular atrophy, and neurodegeneration [159,160,161,162,163].

Quite remarkably, vitamin D can downregulate and suppress NF-kB expression and activity, significantly limiting inflammation. The mechanism whereby the vitamin D/VDR system targets NF-kB includes a physical interaction between VDR and IκB kinase β (IKKβ), which is enhanced by vitamin D, resulting in IκBα stabilization and p65/p50 nuclear translocation impairment [164]. The blockade of p65 translocation leads to the decrease or suppression of NF-kB activation and transcriptional activity, as shown in different cell types stimulated with TNFα—the prototypic cytokine inducing this nuclear factor—not only in immune cells, i.e., DC or B cells, but also in different types of organ resident cells, like human cardiac cells, skeletal muscle cells, thyrocytes, and nucleus pulposus cells [165,166,167,168,169,170].

Thus far, a well-functioning vitamin D/VDR system represents a helpful tool for tackling inflammation and aging, through the downregulation of the TNF-α/NF-kB/p65 signaling cascade in immune system cells and in different tissue cells. This effect seems particularly relevant at the level of striated cells, such as cardiomyocytes or skeletal muscle cells.

### Vitamin D Regulates Heart and Skeletal Muscle Functioning

It is acknowledged that aging per se is a risk factor for heart dysfunction and disease. Age-related cardiac reduced function reflects molecular and cellular modifications, both in non-cardiomyocyte-based components, i.e., vascular cells, fibroblasts, and extracellular matrix, and in cardiac cells, which undergo aberrant processes due to oxidative stress, inflammation, defects in metabolism, cellular repair, telomeres, alteration in gene expression, and post-translational modifications. The dogma that cardiomyocytes are postmitotic cells, terminally differentiated, and unable to undergo cell division, is overcome by the observation that a fraction of cells proliferate and divide in the hearts of the young, adults and the elderly, albeit the cardiomyocyte numbers undeniably decrease with aging, by necrosis [171]. 

The increased number of necrotic cells likely triggers repair-related aberrant inflammation, and a higher chance of autoantigen generation, due to oxidatively modified proteins [172]. Following oxi-inflamm-aging-induced cell death, garbage molecules such as lipoxidation products and advanced glycation end (AGE) products, accumulate from alterations in the protein degradation machinery and autophagy.

Concerning autophagy and apoptosis, it should be mentioned that a functional vitamin D/VDR system—rather than a sufficient vitamin D level alone—can regulate senescence-induced signaling, either by genomic and non-genomic mechanisms, in immunocytes and non-immune cells, retaining the potentiality to be used as a helpful tool against diseases related to inflammation, oxidative stress, or cancer [173,174,175,176].

These processes would impact gene expression toward aging memorization and progress [177]. During cardiac physiological aging, ROS generation is associated with the activation of the inflammasome NOD-like receptor protein 3 (NLRP3), production of biologically active IL-1β and IL-18, and, in turn, pyroptosis, a process causing cell membrane pore formation/rupture and cell death [178,179,180,181].

The activation of NLRP3 is a multistep process, occurring either with or without a priming signal first, denoted as “canonical” or “noncanonical” activation, respectively [182,183,184]. The difference between the canonical/noncanonical mechanisms are exhaustively reported elsewhere, and are beyond the aim of this review.

Independent of the mechanism of activation, an aberrant inflammasome is widely involved in aging, and in an extraordinary number of human age-related diseases. Of note, vitamin D/VDR signaling directly acts as a negative regulator of NLRP3 oligomerization/assembling/activation and IL-1β release, inhibiting pyroptosis [185,186,187].

There is evidence for IL-1β and IL-18 increase from VDR knockout macrophages, activated either through the canonical or noncanonical path [187]. Indeed, ligand-activated VDR attenuates NRLP3 deubiquitination, increasing the level of mitochondrial membrane uncoupling protein-2 (UCP2), which can directly prevent ROS production [188].

Albeit NLRP3 inflammasome activation is known to play a pivotal role in host defense from viral and bacterial infections, dysregulated or excessive activation of the inflammasome is associated with poor outcomes [189]. 

Similar to cardiac cells, skeletal muscle cells show age-dependent biomolecular dysregulation, associated with mitochondrial dysfunction, oxi-inflamm-aging, metabolic disturbance protein breakdown, accumulation of senescent cells, and tissue atrophy [190,191,192].

Age-related decline of skeletal muscle can lead to sarcopenia—not limited to muscle area/mass reduction, but extended to tissue functionality—and frailty [5,193]. In muscle as well, aging-related macroscopic changes reflect biomolecular and cellular modifications. Indeed, tissue changes and deficit are associated with an increase in several inflammatory cytokines, including TNFα, IL-1α/β, IL-6, IL-8, IFNγ, and their soluble receptors (sR) IL-1Ra, TNFαsR, and IL-6sR, referred to as “gerokines”, which altogether constitute the “aging secretome”, likely playing a causal role (rather than being simply markers) in inflammaging [19,194,195]. Even though there is still much to understand about how oxi-inflamm-aging translates into skeletal muscle decline, it is undeniable that the reduction in type II muscle fiber number and size, promoted by TNFα-induced apoptosis, and the unbalance in protein synthesis/degradation, induced by ROS and oxidative stress, merge into functional and mechanical muscle impairment [196,197]. The rise in TNFα or IL-1β increases fat mass toward an unfavorable muscle/fat ratio, which typically characterizes the elderly, especially those with sedentary habits [197]. To date, some intracellular cascades, particularly the TNFα/NF-kB/ubiquitin–proteasome system (UPS) cascade/Akt signaling, play a large part in skeletal muscle inflammaging, representing important targets to counteract skeletal muscle decline. Some strategies, such as protein supplementation, can reduce creatine kinase, but cannot affect circulating gerokines [198,199,200,201]. Another feature of aging muscle is the “differential” resistance to I regarding glucose, protein, and lipid metabolism. Indeed, in the elderly it is common that sensitivity to I concerning glucose is maintained, likely related to reduced utilization of peripheral glucose, while almost simultaneously the so-called “anabolic resistance” to I develops, due to protein synthesis/degradation imbalance (proteostasis loss), a condition preceding clinical manifestations [202,203,204,205,206].

The protective function exerted by vitamin D on skeletal muscle is acknowledged: i.e., vitamin D can directly interfere with and counteract the intracellular paths mediating tissue atrophy, such as proto-oncogene tyrosine-protein kinase Src/extracellular signal-regulated kinases 1 and 2 (ERK1/2)/Akt/forkhead box O3 (FOXO3), a signaling cascade, which upon activation, upregulates atrophic markers such as Atrogin-1 and MuRF1 [207].

An adequate intake of vitamin D can delay the aberrant processes associated with aging and age-related diseases, essentially restoring mitochondrial function and counteracting oxi-inflamm-aging. In myocytes, vitamin D can limit or even neutralize oxidative stress and ROS generation, and the accumulation and expression of garbage molecules like AGE/AGE receptor (RAGE), and high-fat diet-induced I resistance and myosteatosis [208,209,210].

Some experimental and human studies have shown that vitamin D is an effective antioxidant, via activation of the Nrf2-Keap1 antioxidant pathway (the main inducible defense against oxidative stress), showing, e.g., higher capacity than vitamin E to reduce zinc-induced oxidative stress in the central nervous system [211,212,213].

Of interest, in skeletal myocytes, vitamin D anti-aging effects are associated with the inhibition of NF-kB, the crossroad where different oxi-inflamm-aging/senescence paths merge, as previously reported [164,167,214].

Thus far, given the evidence on vitamin D’s role in regulating myocyte metabolism/mitochondrial function/ROS generation, vitamin D deficiency (below 25 nmol/l) is of great interest, particularly in the elderly, since this condition is common among community-dwelling elderly, and very common among institutionalized elderly [67].

To date, it should be mentioned that, while undeniably hypovitaminosis D enhances oxi-inflamm-aging and senescence, an overcorrection of vitamin D status may negatively impact on skeletal muscle cell metabolism, similarly to an overdose of antioxidants. Numerous investigations, indeed, have reported on the potential harmful effects of antioxidant (over)supplementation, especially on muscle fiber formation/muscle regeneration, metabolic homeostasis, and mitochondrial biogenesis [207,215,216,217,218].

## 5. Conclusions

The effect of vitamin D deficiency is classically related to reduced musculo-skeletal functions, and increased risk of disability in locomotion. Besides this undeniable effect, low levels of vitamin D per se strongly affect the aging process, as this molecule regulates cell homeostasis, counteracting oxi-inflamm-aging and cellular senescence with multitargeting actions. In this scenario, the challenge is to restore an adequate vitamin D level.

Some years ago, fortified foods and drinks were introduced, to supplement diet, e.g., some cereals, plant-based beverages like soymilk, orange juice, and some yogurt and cheese. Fortified flour, cornflakes, and juices are more common in the USA, while in Europe, vitamin D enriched margarine, vegetable oil, and milk are more common [219,220,221].

However, from previous simulation studies, fortified foods seem not to be able to provide sufficient vitamin D to the elderly, therefore, individual supplementation is proposed, often in combination with calcium. A supply of about 20 µg vitamin D (800 IU) per day to people over 70, with 50 µg/day recommended as the safe level, and 600 IU, are recommended in subjects from 1 to 70 years of age. Supplementation with 20 µg vitamin D and 1000/1200 mg calcium increases the vitamin D level, suppresses secondary hyperparathyroidism (lowering the parathyroid hormone/PTH), and improves bone and muscle strength [67]. As found in several studies aimed at reducing osteoporotic fractures in the elderly, a lower amount of vitamin D is required (10 µg/day) in the presence of calcium supplementation (1000 mg) [222]. 

To date, research studies have explored the efficacy of vitamin D supplementation in the prevention and treatment of chronic conditions of aging, nevertheless the translation to standardized application is lacking, for a number of reasons. There is still an urgent need of clear indications as to what is the best dosage for a supplement, considering, e.g., the comorbidities. Moreover, the lack of a universally accepted standard for vitamin D assessment, i.e., due to variability in methods, strictly connects with the absence of clarity in vitamin D status definition (sufficiency/insufficiency/deficiency), based on a standardized reference range. Plasma 25(OH)D (the more stable analyte) is currently used to assess vitamin D status, but it is recognized that the circulating level of this metabolite varies depending upon season, latitude, clothing, dietary habits, race, pigmentation, skin thickness, sex, and age [67]. All the concerns about vitamin D determination and supplementation are exhaustively reported in the latest consensus statement from the 2nd International Conference on Controversies in Vitamin D [223].

So far, the role of vitamin D status in immunosenescence, inflammaging, and whole-body aging is based on scientifically documented data, and is fully recognized by the scientific literature, but still there are important limitations to translate knowledge into clinical practice, with important medical and socioeconomic consequences, considering the high and growing number of individuals aged 65 and older.

The DO-HEALTH multicenter clinical trial, currently running in 2157 community-dwelling European men and women aged 70 and older, combines vitamin D (2000 IU/day) treatment with omega-3 fatty acids intake (1000 mg/day), and a 30-min physical activity (3 times/week home exercise). This trial addresses several health domains (cardiovascular, muscle, bone, brain, and immunity) and hopefully will help to implement clinical practice [224].

This review wants to focus as much attention as possible on these aspects, even though some limitations, such as lack of discussion on sex-dependent variability, or differences between active and sedentary elderly, are present.

Given the importance of this topic, more accurate basic and clinical human research is necessary to approach vitamin D status determination, opening the way to future scenarios of personalized treatment.

## Figures and Tables

**Figure 1 ijms-24-04546-f001:**
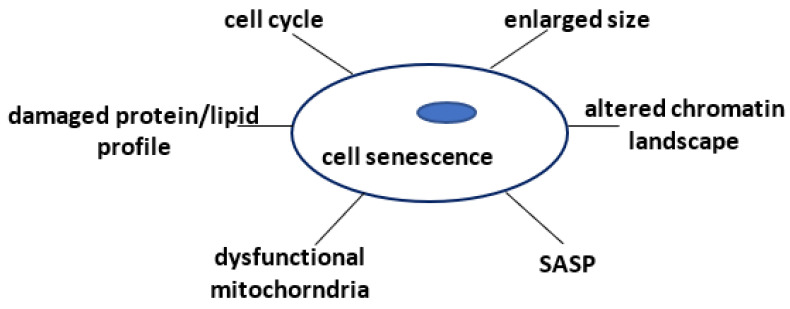
The main cellular modification induced by senescence. Aging induced modifications at the cellular level include morphological, enzymatic, and functional variations.

**Figure 2 ijms-24-04546-f002:**
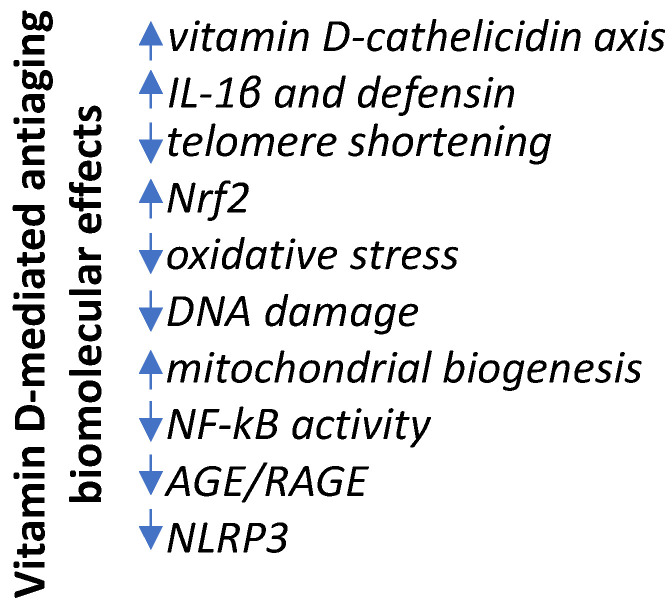
The molecular mechanisms involved in the anti-aging effects of vitamin D. An adequate level of vitamin D counteracts age-dependent changes at multi-target biomolecular levels, ending in effects against infections, inflammation, oxidative stress, garbage molecule accumulation, and aberrant mitochondria functioning.

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
