# Peer review of "Vitamin D as a Shield against Aging"

_ijms, 2023, doi:10.3390/ijms24054546_

Round 1

Reviewer 1 Report

 This is an interesting and in-depth review focusing on the role of vitamin D in geroprotection based on its ability to shape cellular/intracellular processes and drive the immune response toward immune protection, against infections and age-related diseases.

I would suggest some insights.

Was recently reported that IL-33 results in immunosuppression by inducing thymic involution-associated naive T cell dysfunction with aberrant expression of aging-associated genes and also suggest that targeting IL-33 or ST2 is a promising strategy to rejuvenate T cell immunity (PMID: 36371464). Furthermore, the IL-33:ST2 axis seems to be a promising avenue to explore in attempts to address the age-related sarcopenia and associated defects in muscle repair subsequent to injury or atrophy,  a major health problem with our aging population structure, exerting a strong impact on mobility, independence and quality of life (PMID: 26872699). Emerging evidence suggests a functional link between vitamin D and the IL-33/ST2 axis, which acts through hormonal influences and immune-mediated effects, as well as cellular and metabolic functions: their pathogenetic crosstalk in inflammatory conditions may be hypothesized (PMID: 33752232, PMID: 33488605, PMID: 35514132 ).

Another aspect concerns apoptosis or programmed cell death that plays a central role in the ageing process (PMID: 17584051) and the role of vitamin D and its receptor regulating autophagy signaling (PMID: 34403722) and apoptosis (PMID: 34546851, PMID: 36588053, PMID: 36159807).

Figure 1 mention “infections, cancer and autoimmunity” and a paragraph refers to infections. Adding a short paragraph regarding autoimmunity would also be interesting.

Author Response

We thank the R1 for the constructive criticism.

  • The role of IL-33 and ST2 related to aging and to vitamin D interplay has been addressed in the revised manuscript, subparagraph 2.2, page 6, lines 190-196 and in subparagraph 3.2, page 10, lines 378-383, respectively. The suggested references have been quoted and listed as new number 43, 44, 117-119.

  • A comment on the regulation of apoptosis and autophagy by vitamin D/VDR system has been addressed in the revised manuscript, subparagraph 4.1, page 13, lines 518-522; the suggested references have been quoted and listed as new numbers 173-176.

  • Topics on autoimmune disease and vitamin D have been addressed in subparagraph 3.2, page 10, lines 369-383, and related references have been quoted and listed as new numbers 114-119.

Reviewer 2 Report

The manuscript, entitled " Vitamin D as a shield against aging " This work is merited for publication in International Journal of Molecular Sciences after some minor modification. So, I have some points that may help to improve the work as follows:

1-Abstract is good but need more explain about the main aim of work

2- The introduction should be extended to discuss the hypothesis and research questions in details. Additionally, the introduction should cover the recent literature related to this subject.

3- The conclusion

A section for conclusions need more explain and should include the most significant findings and future works only.

Line 166-225, please rephrase it.

Line 258-268, please rephrase it.

Line 464-482, please rephrase it.

4- English writing should be checked by a native English-speaking expert.

Author Response

We are grateful to the R2 for the important suggestions.

  1. We have added just a sentence “To this aim, the main biomolecular paths underlying immunosenescence and inflammaging are revised as potential biotargets of vitamin D” within the aim of the Abstract, line 26-28, in order to better highlight the main aim of the review. We apologize, we could not add more due to word limit.
  2. The Introduction has been modified and extended, following the Reviewer’s suggestion, page 3, lines 82-90. Recent literature has been quoted and listed as new numbers 6-14.
  3. The Conclusion has been focused on significant issues and future works. The sentences removed from this paragraph have been reallocated in subparagraph 3 and modified as convenient. The reference list has been updated (224 references in the revised manuscript).

Lines 166-225 in the original manuscript, corresponding to subparagraph 2.2, have been rephrased, as requested.

Lines 258-268 in the original manuscript (beginning of subparagraph 3.1) have been rephrased.

Lines 464-482 in the original manuscript (subparagraph 4.1) have been rephrased.

  1. English throughout the text has been revised by native-speaking expert.

Round 2

Reviewer 1 Report

A review that takes stock of the role of vitamin D in aging in a clear and stimulating way

Reviewer 2 Report

Authors have suitably revised the manuscript by addressing the reviewer comments and suggestions. This can be accepted for publication.